# Digital cloning of online social networks for language-sensitive agent-based modeling of misinformation spread

**Prateek Puri**[ORCID]*, **Gabriel Hassler, Sai Katragadda, Anton Shenk**

RAND Corporation, Santa Monica, California, United States of America

* ppuri@rand.org

**Data Availability Statement:** Raw X data cannot be shared publicly due to data sharing agreements within the X and Brandwatch platforms. However, all data extracted from our simulations are available

## Abstract

We develop a simulation framework for studying misinformation spread within online social networks that blends agent-based modeling and natural language processing techniques. While many other agent-based simulations exist in this space, questions over their fidelity and generalization to existing networks in part hinder their ability to drive policy-relevant decision making. To partially address these concerns, we create a 'digital clone' of a known misinformation sharing network by downloading social media histories for over ten thousand of its users. We parse these histories to both extract the structure of the network and model the nuanced ways in which information is shared and spread among its members. Unlike many other agent-based methods in this space, information sharing between users in our framework is sensitive to topic of discussion, user preferences, and online community dynamics. To evaluate the fidelity of our method, we seed our cloned network with a set of posts recorded in the base network and compare propagation dynamics between the two, observing reasonable agreement across the twin networks over a variety of metrics. Lastly, we explore how the cloned network may serve as a flexible, low-cost testbed for misinformation countermeasure evaluation and red teaming analysis. We hope the tools explored here augment existing efforts in the space and unlock new opportunities for misinformation countermeasure evaluation, a field that may become increasingly important to consider with the anticipated rise of misinformation campaigns fueled by generative artificial intelligence.

## Introduction

Online misinformation has played a critical role in shaping public opinion on national issues such as election security [1, 2], vaccine effectiveness [3, 4], climate science [5, 6], and many other topics in recent years. As social media platforms continue to proliferate in volume [7] and as technologies such as generative artificial intelligence (AI) mature, misinformation campaigns are expected to increase in both severity and scale [8, 9]. Consequently, significant effort has been focused on developing strategies to understand misinformation spread [10, 11] and design mitigation strategies [12–14]. Within many of these frameworks, misinformation spread is viewed through the lens of network theory and infectious disease modeling [15, 16], whereby infected social network nodes (misinformation spreaders) expose node neighbors

form the Harvard Dataverse database (https://doi.org/10.7910/DVN/O17AWX). https://doi.org/10.7910/DVN/O17AWX.

**Funding:** P.P. received funding for this work. This work was funded by an internal grant within The RAND Corporation (Grant Number: IVIRP23004). The RAND Corporation is non-profit bipartisan think tank whose mission is to support public well-being (https://www.rand.org/). The grant was intended to support the research and development of novel tools/methods that have applicability to policy analysis and furthering the public good. The sponsors did not play any role in the study design, data collection and analysis, decision to publish, or preparation of the manuscript.

**Competing interests:** The authors have declared that no competing interests exist.

(social media connections) to infection, thereby inducing further infections. Consequently, many proposed misinformation countermeasure strategies are rooted in public health concepts such as inoculation via media literacy training [17], quarantining of infected individuals via account blocking [18], inoculation via fact-checking [19], and others.

While mitigation strategies have been evaluated in randomized control trials [20–22], it is difficult to anticipate how their effectiveness may change when applied at scale under rapidly shifting online landscapes. A growing body of research is leveraging agent-based modeling (ABM) to explore countermeasure evaluation [23–27] in low-cost, flexible environments. Such systems allow for the simulation of misinformation campaigns across synthetic networks that are customizable in both structure and scale. While still subject to the typical limitations of agent-based models [28], such as computational complexity and explainability, these platforms allow for probing of more granular dynamics than typically available via alternative computational techniques [29]. However, a majority of agent-based misinformation infection models rely on infection probabilities that are static for each user and for each topic of misinformation that is explored. In reality, the likelihood of information spread between social media users has a complex relationship to user preferences, user community, and the topic being discussed [30, 31]. The lack of such dynamism in static infection models limits investigation of how countermeasure effectiveness varies in response to these variables.

To address these concerns, in this mixed methods article we augment existing ABM frameworks with machine learning (ML) methods to generate infection pathways that are sensitive to user community, user preferences, and topic of discussion. A known misinformation-spreading network is 'digitally cloned' by downloading X (formerly Twitter) activity histories for each user within the network, which are further processed to train ML models to produce user-specific infection probabilities. Secondly, we introduce an information mutation feature into our ABM that leverages large language models (LLMs) to predict how information morphs as it is transmitted through a network. We evaluate our framework, which includes both infection and mutation models, by seeding the cloned network with a sample of recorded posts within our base network and comparing propagation dynamics between the two. Lastly, we build our system predominantly in Julia, a programming language which may offer scaling advantages when simulating dynamics in larger, and more realistic, networks.

Put together, this work presents progress towards building systems to (1) better evaluate online misinformation countermeasures in low-cost environments and (2) perform red team analysis on what linguistic framing and/or discussion topics render online networks most vulnerable to misinformation spread. In the following sections, we outline our method, describe our results, and summarize future steps for this research.

## Materials and methods

### Misinformation event selection

Cloning all users within a social media platform is not computationally feasible, nor necessary, given the aims of this work. Consequently, the first step in creating a digital clone is identifying a relevant social media subnetwork. Ideally, such a subnetwork would consist of highly connected users who regularly share misinformation posts amongst one another, as such a network is likely to exhibit rich propagation dynamics for our ABM to replicate. However, identifying such a subnetwork, and evaluating its properties, is a non-trivial task. Instead, we focused on the less burdensome task of identifying a viral misinformation post authored by a given user and then backtracking a subnetwork by identifying users who interacted with this post. While a subnetwork identified via this route may not be optimally structured, it was sufficient for many purposes of this work, as will be discussed. Network backtracking will be

described further in the Network Selection section; in this section we focus on the selection of a viral source post ($T_S$) submitted by a source user ($u_S$).

To narrow our consideration pool for $T_S$, we focused on a set of X posts flagged in a COVID-19 vaccine hesitancy dataset established in the literature [32]. We selected this dataset both for its robustness and for its relevance to recent misinformation conversations. Within this dataset, we restricted our search to events that occurred in 2021 to avoid data volatility in the period surrounding the initial onset of the COVID-19 pandemic.

Within this narrowed set of events, we randomly sampled a set of posts and leveraged the X application programming interface (API) and rank them in descending order of retweet (RT) count. We hand-evaluated the top ten results and selected a post related to vaccine conspiracy theories authored in May 2021 that generated a total of ~600 retweets, placing the post in the ~90% percentile in terms of retweet activity [33]. The tweet was chosen for its linguistic coherence and relative self-containment compared to the other reviewed posts. We do not provide the text of the source post here to protect individual privacy.

## Network selection

The next step was to construct a network of users who engaged with $T_S$, or were connected to such users, to serve as a foundational subnetwork for our cloned ABM. We leveraged the Brandwatch [34] platform, a third-party collector and distributor of social media datasets, to track the set of users, $U_T$, who shared $T_S$ or any subsequent retweet of $T_S$. We then derived a subnetwork consisting of these nodes and a modified set of their immediate one-hop neighbors. For the remainder of the article, we will define the following terms: if a user $u_i$ follows a user $u_j$, $u_i$ is a **follower** of $u_j$ and $u_j$ is a **followee** of $u_i$.

In more detail, for each user, $u_t$ within $U_T$, we downloaded tweets posted between February 2021 –April 2021 that were either (1) retweeted by $u_t$ or (2) posted by $u_t$ and later retweeted by another X user. This period, which precedes $T_S$ by three months, was chosen to probe network relationships/behavior that existed in the timeframe immediately prior to $T_S$. The set of all users present in this dataset, either as a retweeter or original poster, is denoted as $U_A$. Bidirectional edge relationships between users in $U_A$ were defined as:

$$e_{ij} = \begin{cases} 1 \ if \ |R_{ij}| > 0 \\ 0 \ if \ |R_{ij}| = 0 \end{cases} \tag{1}$$

where $e_{ij}$ is a binary variable that indicates whether an edge relationship between $u_i \rightarrow u_j$ exists, $R_{ij}$ is the set of posts authored by $u_i$ and subsequently retweeted by $u_j$, and $|R_{ij}|$ is the size of this set. To make our network size manageable for running simulations given available resources, we further narrowed the network via the following process. Firstly, we define $N_A$ as the subnetwork to be used within our ABM and initially set $N_A = U_T$. We consider all users in $U_T$, but not present in $N_A$, and rank each user in this set, $u_i$, according to the number of incoming $\left( \sum_{j \epsilon N_A} |R_{ji}| \right)$ and outgoing $\left( \sum_{j \epsilon N_A} |R_{ij}| \right)$ posts they've participated in with users within $N_A$, weighing each equally. Following a modified snowball sampling procedure, we add the highest ranked user to $N_A$ and repeat the process iteratively until we have 10,000 users in $N_A$ (S1 Appendix in S1 File). Considering both the in-degree and out-degree connections of each user to $N_A$ during the subnetwork selection process helps balance including users who source information with those who disseminate information.

One-hop nearest-neighbor and snowball sampling have been known to produce subnetworks that differ from their global networks across metrics such as centrality, average path length, and others [35, 36]. The sampling procedure ensured here is deemed adequate but not

optimal. While we focus the remainder of our analysis on the ability of our ABM to replicate dynamics within $N_A$, we nosste that in future studies, alternative sampling techniques may be employed to generate ABM subnetworks with properties more representative of social networks of interest.

Lastly, on the X platform, each profile is associated with a set of users who follow the account and a second set of users the user of the account themselves follow. However, these relationships do not fully capture information spreading pathways. We infer edges between nodes through Eq 1 rather than extracting followee → follower relationships from the X API for the two following reasons:

1. Users are capable of retweeting information from individuals they do not follow. These information pathways are captured via the method above but are not captured by solely examining a user's followees

2. When this research was conducted, the X API has rate limits that would make such processing infeasible for our network

For the remainder of the text, we will define follower and followee relationships as edge relationships determined by Eq 1 rather than those stated on a user's X profile.

## Community detection

With $N_A$ defined, we performed Leiden community detection [37] to segment each user into a community, allowing community-community interactions to be studied within our ABM. This process yielded nine total communities. Visualizations of these communities, as well as interactions between them, are presented in Fig 1A. In the figure, each node represents a community, where the node size (edge thickness) is proportional to the community size (number of total edges between users from separate communities). A visualization of the network structure within an example community ('Free Assange') is presented in Fig 1B, and the degree distribution for $N_A$ is shown in Fig 1C. Community labels were extracted by leveraging the BERTopic [38] library to apply a class-based term-frequency inverse-document-frequency (c-TF-IDF) technique to a random sample of ~10,000 tweets from each community (S1 Appendix in S1 File).

## Data extraction

We segment the Brandwatch historical X data pulled for each user in $N_A$ into three timeframes as follows:

- Period I (Feb. 01, 2021 –Mar. 31, 2021)

- Period II (April 1, 2021 –April 15, 2021)

- Period III (April 16, 2021 –July 31, 2021)

Data from Periods I-II are leveraged to establish network relationships, extract user features needed for the ABM, and train both the infection model and the mutation model. Period III data is leveraged to evaluate both our infection model and our mutation model as well as to evaluate the performance of our ABM. A notional diagram of the roles these time periods play in our pipeline is displayed in Fig 2.

## Ethics statement

We only extracted public posts which users, by agreeing to X's data privacy terms and conditions, agreed to make broadly publicly accessible. The RAND Corporation Human Subjects

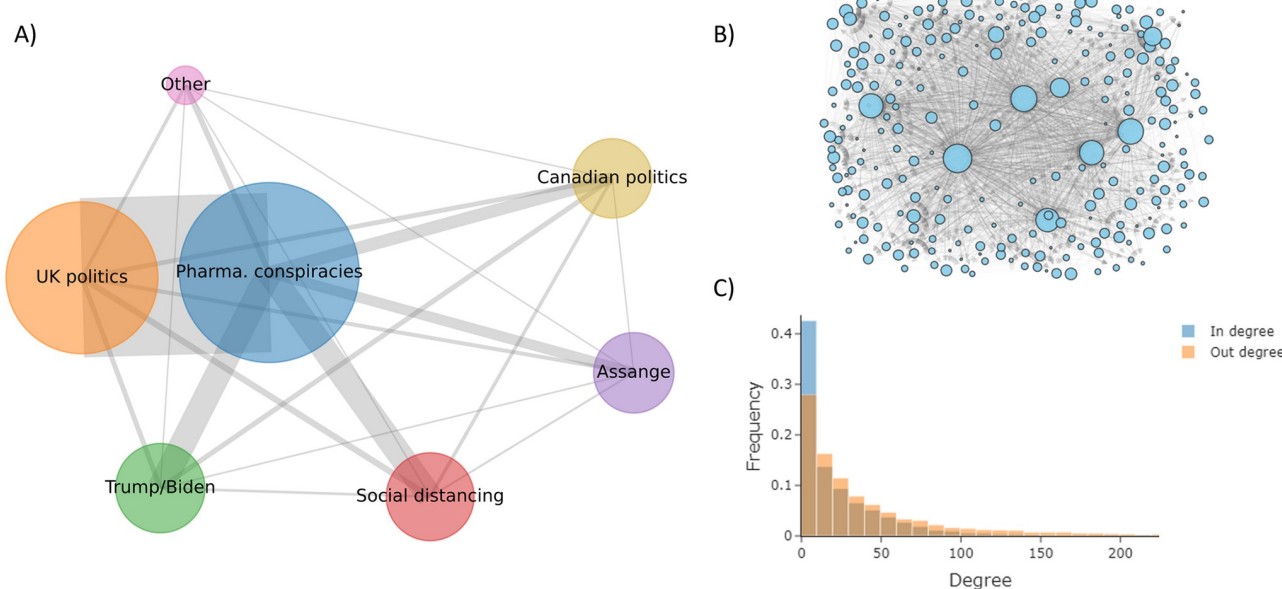

**Fig 1. Base network characterization.** (A) Network diagram of our base social media network. Node size is proportional to community population number, and edge thickness is proportional to the number of user edges between two community nodes. The labels are extracted by applying topic modeling to recorded tweet history within each community. (B) A directed network diagram for a sample of users within the 'Assange' community where each node represents a user within the community, node size is proportional to follower count, and edge transparency is proportional to node out-degree (C) The in-degree and out-degree distribution of our base network.

Protection Committee (HSPC) reviewed and approved the data collection and handling protocols within this project. Given no private posts were obtained during this work, and given X's data privacy policy, the HSPC board determined additional consent was not required from the studied users. To minimize the amount of personally identifiable information ingested by our system, only data fields such as user id, post text, post engagement type (post, reshare, reply, etc.), and number of followers/followees were analyzed within our research. However, a user id can be linked to a user's profile, where, in certain cases, users may opt to share personally identifiable information publicly. Additionally, Brandwatch's policies only permit authorized

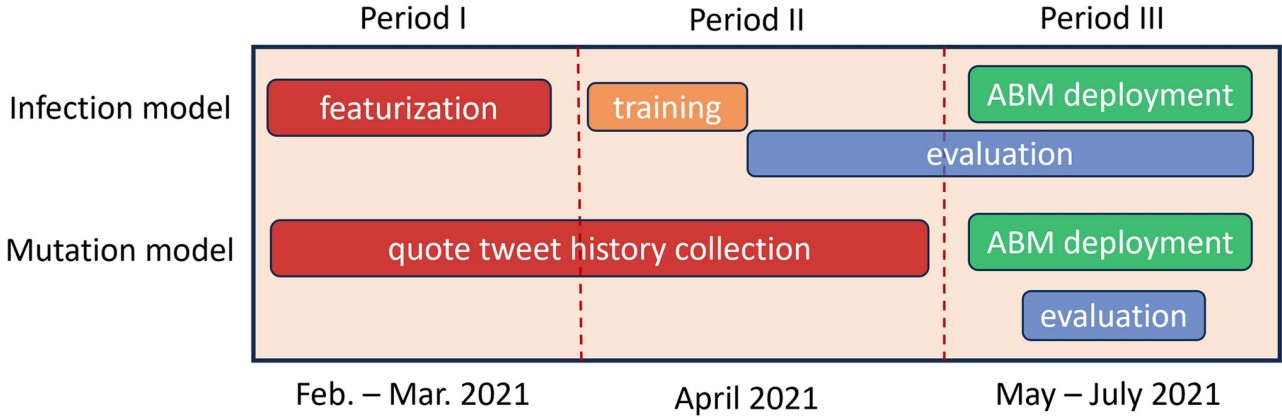

**Fig 2. Data segmentation.** Diagram displaying how historical social media data from users in our base network is distributed amongst various stages of development stages for the ABM, infection model, and mutation model.

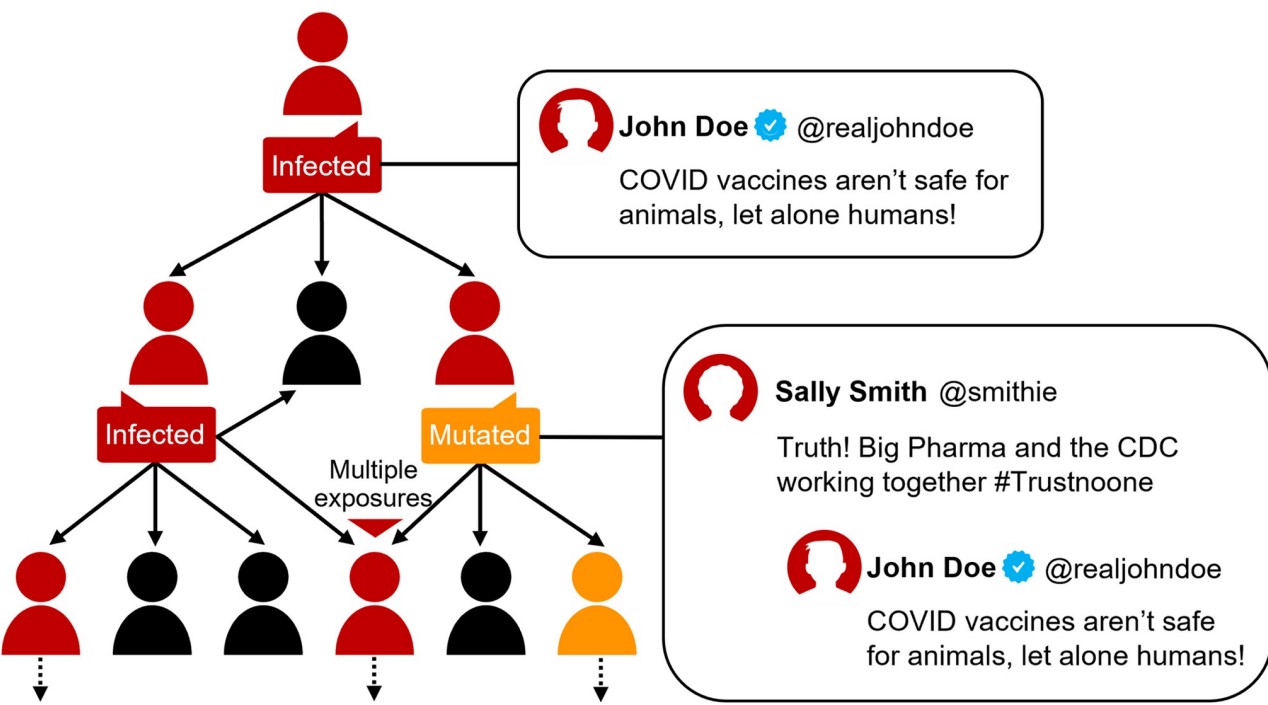

**Fig 3. Schematic diagram of the ABM logic.** Illustrative diagram conveying the operating principle behind the ABM. A source user is infected when they share a source post. Their followers are exposed to their infection, some of which will become infected themselves by resharing the source post. This process continues across infection layers, with a fraction mutating the infection as they transmit it by adding additional commentary to their reshare post.

users to access data extracted on their platform. For both reasons, we do not release any of the raw social media data we analyzed in this work. However, we do present the raw data from our anonymized ABM simulations within a public data repository.

## ABM dynamics

We build an agent-based susceptible-exposed-infective (SEI) model where individuals can either be susceptible ($S$, have not been infected), exposed ($E$, have been infected by misinformation but have not yet retweeted misinformation), or infective ($I$, have retweeted misinformation). A detailed workflow diagram of the ABM logic is displayed in Fig 3, and a condensed summary is provided as follows:

```
SEI Model Pseudocode
I. source author uS is exposed to tweet TS
  • set the state of user uS to exposed: S(uS) → E
  • initialize the set of exposed users: SE → {uS}
  • set author uS infection time: tS → 0
  • set the originator of tweet: Orig(TS) → uS
II. while |SE| > 0:
  • Find the user ui with the lowest infection time ti
  • user ui is infective
    • S(ui) → I
  • remove user ui from the set of exposed users
    • SE → SE \ ui
  • for each susceptible follower uj of ui (i.e., all uj such that uj
    follows ui and S(uj) = S):
```

- compute infection probability as $IP \rightarrow IM(u_j, Orig(T_i), T_i)$ ($Orig(T_i)$ is the originator of tweet $T_i$)
- sample a uniform random variable: $x_{IP} \sim Uniform(0, 1)$
- if $x_{IP} \leq IP$:
  - follower $u_j$ is exposed: $S(u_j) \rightarrow E$
  - follower $u_j$ is added to the set of exposed users: $S_E \rightarrow S_E \cup \{u_j\}$
  - follower is assigned an infection time
    - sample $\Delta \sim Exponential(1)$
    - $t_j \rightarrow t_i + \Delta$
  - compute quote tweet probability: $QP \rightarrow QM(u_j)$
    - $QM(u_j)$ is the empirical frequency at which user $u_j$ quote tweets (as opposed to retweets) from their observed twitter history. This quantity is pre-computed for each user.
  - sample $x_{QP} \sim Uniform(0, 1)$
  - if $x_{QP} \leq QP$:
    - $T_j$ generated by LLM
    - $Orig(T_j) \rightarrow u_j$
  - if $x_{QP} > QP$:
    - $T_j \rightarrow T_i$
    - $Orig(T_j) \rightarrow Orig(T_i)$
- if $x_{IP} > IP$:
  - *continue*

where $S(u_j)$ represents the SEI state of user $u_j$; $S$ is the susceptible state; $E$ in the exposed state; $I$ in the infective state; $IM(u_j, u_k, T_i)$ is a function that returns the probability of infection with features derived from the follower $u_j$, tweet's source author $u_k$, and the tweet itself $T_i$ (see Infection model section below). One thousand iterations of the above process are executed for each explored ABM scenario to capture stochastic variation.

## Infection model

The infection model estimates the probability $IP = IM(f_j, u_k, T_i)$ that a particular follower $f_j$ of user $u_i$ will retweet tweet $T_i$, originally posted by user $u_k$. To provide features for this model, we calculate vector embeddings for $T_i$ and also provide the following set of information extracted from $u_k$ and $f_j$ during Period I: the number of followers, the number of followees, the follower-to-followee ratio, the frequency at which their tweets were retweeted, the frequency at which they retweeted followee tweets, and a set of embeddings extracted from their retweet history (Fig 4). A vector is constructed from all non-embeddings features and concatenated with the embeddings vectors to form a final set of model inputs.

As noted above, there are two types of embeddings ingested by the model: a set (user-level) calculated for $u_k$ and $f_j$ and another set (tweet-level) extracted from $T_i$. For the user-level set, we generate 384-dimensional embeddings for each Period I post that is either authored by $u_k$ or reshared by $f_j$ using the all-MiniLM-L6-v2 model in the sentence-transformers Python package [39]. We use an autoencoder to further reduce the embedding dimension to 24 and then average these reduced embeddings for each user, generating a tweet embedding vector for $u_k$ and retweet embedding vector for $f_j$. The $u_k$ and $f_j$ embeddings provide information on the type of content each user has historically posted and reshared, respectively.

For the tweet-level embeddings, we apply all-MiniLM-L6-v2 to $T_i$ as above but use a separate autoencoder to reduce the embedding dimension to 96. We concatenate all three sets of embeddings mentioned above into a vector that is ingested, along with the non-embeddings features, by the model. By providing both tweet-level and user-level embeddings, we enable the model to parse how the topic of a given post relates to historical user preferences. Here, we chose a greater dimension for the $T_i$ embeddings than the user-level embeddings so that the

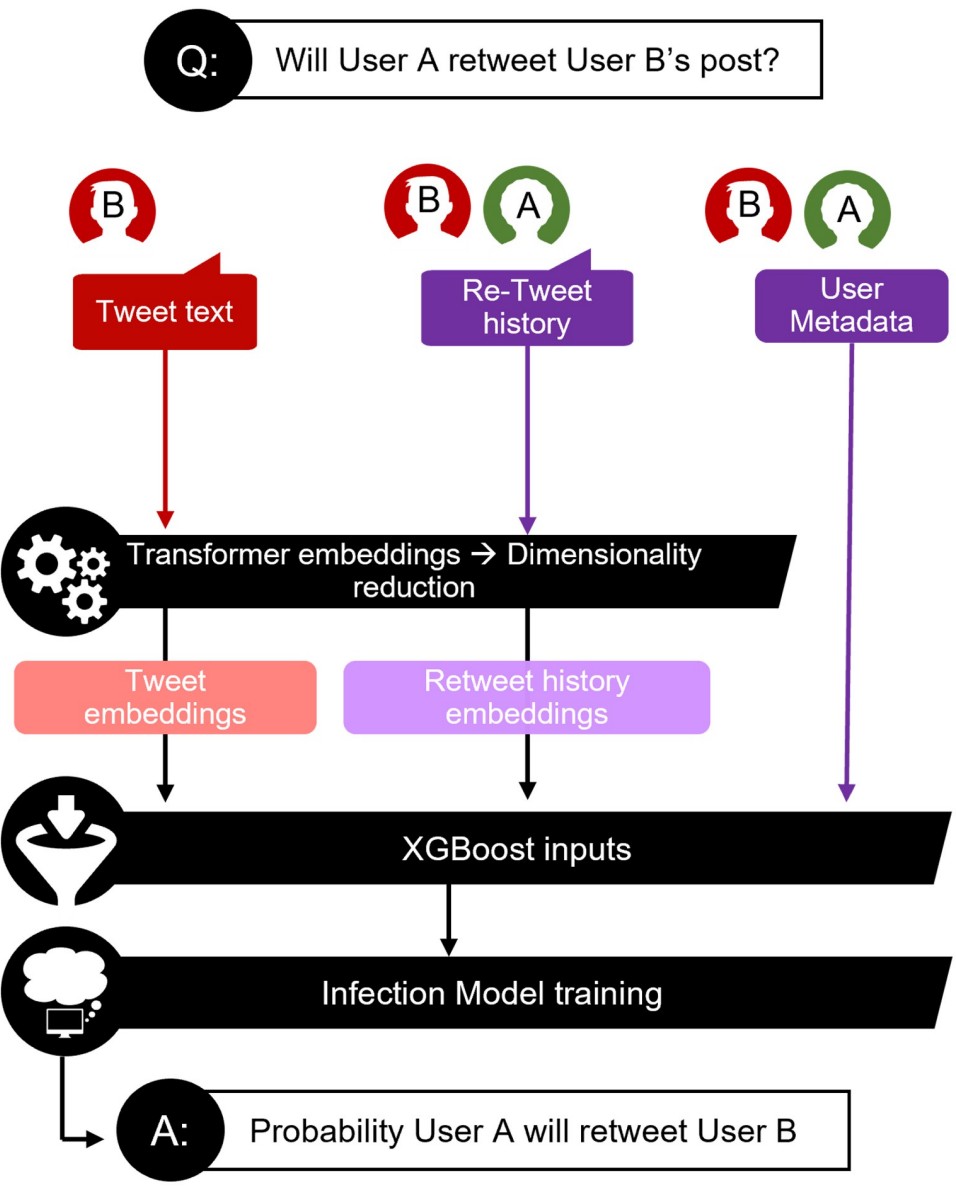

**Fig 4. Schematic diagram of the infection model training process.** Diagram describing the training process for the infection model, which predicts whether User A will retweet User B's post. The core model is a gradient boosted classifier with three sets of input features (i) transformer embeddings of User B's post (i) transformer embeddings extracted from both historical tweets User B *has authored* and historical tweets User A has retweeted *from* others (iii) user metadata—such as number of followers, number of followees, etc.–from both User A and User B. Once the infection model is trained, it can be deployed to estimate the likelihood of infection spread.

infection model would be more sensitive to the text of the tweet spreading through the network.

After pre-processing the data, we trained a gradient-boosted tree classification model using the EvoTrees Julia package [40] to compute the probability that a follower will retweet a particular tweet from a particular followee. The data in our training period included 35,330,188 tweets, with a total of 130,432 retweets (0.37% overall retweet rate). Here, we assume all followers of a user are exposed to their posts, meaning a lack of reshare between a user and their follower will be labeled as a negative event within our binary classification training set. We

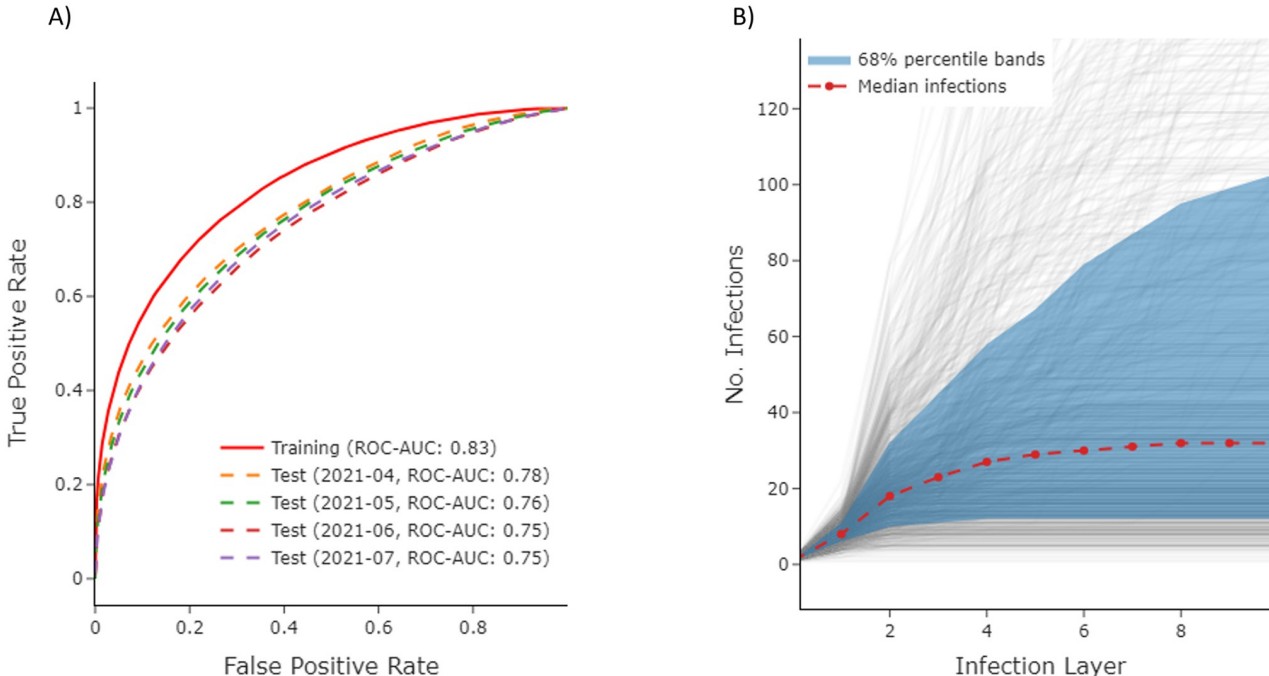

**Fig 5. Infection model and ABM characterization.** (A) The AUC-ROC curves for the infection model across the training set and set of hold-out test sets from different time periods that occurred after all recorded training set events. Slight overfitting between the training and test sets is observed; however, performance across test sets appears roughly consistent, suggesting Period I and II user behavior encoded during the training process is indicative of forward-looking information sharing behavior for multiple months. (B) The number of infections across infection layers for a set of ABM trials for a sample source post. The grey lines represent traces obtained from each of the 1000 trials. The blue bands denote the 68% percentile bands across these trials, with the red dashed line representing the median number of infections at each infection layer across all trials.

partitioned the data into a training set of roughly 20% of observations and a test set of the remaining 80%. We used the hyperopt Python package [41] for identifying optimal hyper-parameters subsequently used for fitting the final model.

We evaluate our model on a set of four Period II-III test sets, each consisting of samples taken from each month in the April 2021 –July 2021 time frame. We observe a degree of over-fitting between the training and test sets; however, we notice only very slight performance deg-radation across time, suggesting Period I-II user behavior encoded during the training process remains relevant to user information sharing tendencies for multiple months (Fig 5A). The persistence of infection model performance bodes well for the maintainability of simulation frameworks leveraging the model. For example, if performance degraded sharply over time, the model would need to be retrained frequently to produce infection probabilities aligned with current user preferences, imposing significant model maintenance costs. The lack of such degradation implies an infection model, once trained, may produce reasonably accurate infec-tion probabilities across a time horizon spanning several months.

Because the boosted tree model involved regularization, its outputs did not correspond per-fectly to empirical probabilities and had to be recalibrated to conform to actual probabilities. To recalibrate tree model outputs, we binned the prediction from each observation in the test dataset by quantile (100 quantiles total). We then calculated the empirical probability of a retweet among all observations in each quantile. Finally, to smooth the calibration curve, we fit a degree-11 polynomial with non-negative coefficients to the calibration curve, which we used to adjust any boosted tree model outputs for the simulation model.

## Mutation model

Rather than remaining static, misinformation often gets mutated as it travels through a social network, as users interpret and transmit information through their own unique lens. On the X platform, users can add custom commentary to posts they retweet from other users, with such posts often garnering more attention than standard reshares. For example, within our Period I-II dataset, these so-called 'quote tweet' (QT) events experienced an average of ~50% more impressions than standard retweet events, as measured by BrandWatch's monitoring metrics [42, 43].

While previous work has highlighted the importance of information mutation to misinformation propagation dynamics [44, 45], such mutations are difficult to model, posing challenges to incorporating them into ABMs. In this work, we explore how LLMs may be leveraged to reduce this capability gap.

The anatomy of a quote tweet event consists of a parent tweet a user shares (PT, i.e., *'Climate scientists lie AGAIN about impact of fossil fuels on sea levels'*) and additional commentary the user adds to the PT (AC, i.e., *'First climate scientists, now vaccine scientists. . . #NoTrust'*). Upon authoring of the QT, followers of a user will see an aggregated post consisting of AC + PT concatenated together (i.e., *'First climate scientists, now vaccine scientists. . . #NoTrust*: *Climate scientists lie AGAIN about impact of fossil fuels on sea levels'*).

Our mutation model is described in depth in S2 Appendix in S1 File, and a high-level overview is provided here. For a subset of users, we instructed the gpt-3.5-turbo model to predict user AC given a PT for a set of Period III QT evaluation events, sampling from the user's Period I-II QT history to provide few-shot prompting context. We only selected users who had at least 25 QTs in Period I-II and 20 QTs in Period III for mutation modeling to ensure we had enough QT events for context building and model evaluation, respectively. Further, the mutation model predicts the text of a given QT event but not whether it will occur. For modeling the latter, a random draw based on a users' Period I-II QT:RT frequency count ratio determines whether a user exposes his followers to a mutated (QT) or un-mutated (RT) strain of their infection within our ABM (Fig 3).

To evaluate the quality of the QT predictions, we computed cosine similarities between the embeddings of the LLM prediction and the ground truth text. Amongst the set of selected users, we observed an average cosine similarity of 0.54 between embeddings of the LLM ACs and ground truth ACs (S2 Appendix in S1 File).

While the data filters mentioned above limited the mutation model user set to ~1% of total $N_A$ users, in the future, increasing the length of Period I-II, exploring longer context window models, and additional prompt engineering may improve results even further. Due to the limited user set, our mutation model exerted minor influence on our ABM outputs (<1% difference in infection rates compared to neglecting mutations); however, this trend is expected to change as the capability is expanded to more users. The prototype method explored here presents a step towards modeling more complex online misinformation behavior through LLMs and simulating information sharing not solely restricted to reposts.

## ABM runtime

The runtime of the ABM is determined by the number of mutation events, the average infection probability, and the degree distribution of the network. For each tweet, we run 1,000 simulations to accurately capture uncertainty in the infection dynamics. When allowing for mutations, the runtime for 1,000 simulations is ~5 minutes. In this case, OpenAI API calls were run serially with an average response time of 1.13 seconds and accounted for ~70% of total run time. An equivalent model without mutations required only ~20 seconds of runtime

for 1,000 trials. Note that the non-mutation model benefits from both avoiding OpenAI API calls and the ability to pre-compute all required infection probabilities prior to running the ABM given the static infection tweet text. Infection probabilities for mutations, which are not known *a priori*, cannot be pre-computed in this way. However, parallelization of OpenAI calls and increasing parallelization of ABM trials can reduce run times further. Assuming conservative $\sim N^2$ scaling of computation time with network size, simulating networks of order $\sim$1M users may be feasible.

## Results

After establishing our cloned network and infection model, we conducted benchmark tests to evaluate its performance. Firstly, we seeded the synthetic network with $T_S$ as discussed in the Misinformation event selection section and monitored propagation dynamics over 1,000 trials. The distribution of infection number across all simulated trials, displayed as a function of infection layer, is shown in Fig 5B.

Direct comparison of both the total infection number and total infection rate (infection number / exposed users) between the cloned and base networks is complicated due to their different sizes. For example, $u_S$ has $\sim$100,000 followers, while $N_A$ only possesses $\sim$10,000 users total. While $N_A$ contains users infected in the base network, it does not contain all users *that could have been infected*. Put another way, the observed outcome in our base network is one sample drawn from possible outcomes that could be observed if one were able to initialize identical versions of the base network prior to applying $T_S$. Since our ABM does not contain the same set of users, it cannot sample the full outcome space available to our base network and produce directly comparable infection numbers.

To account for the difference in network sizes, for all work presented below, we multiply infection probabilities by a constant factor α. We explored a range of values and found that α = 3.0 resulted in total infection numbers in our cloned network similar to that observed in the total network.

As an alternative to comparing direct infection numbers, we explore how well our ABM anticipates variations in virality amongst posts by seeding our network with both

(i) a set of $\sim$10,000 Period III posts sampled across all users in $N_A$

(ii) a set of $\sim$1000 Period III posts sampled from $u_S$

For posts within both (i) and (ii), we extracted the number of infected users for each post through our Brandwatch dataset and compared the resulting value to that obtained through our ABM. The comparison of (i) helps assess how well the ABM can predict variations in virality amongst a set of posts by considering differences in both user-level features and post text. On the other hand, the comparison of (ii) helps isolate the degree to which the ABM can anticipate how differences in post text impact virality. Due to computational requirements of running such a large volume of simulations, we truncate each ABM trial after the first infection layer. For (i), we also normalize infection number by the number of post author followers to set a consistent scale across observations. Lastly, since events are randomly sampled from each user's post history, not all posts with (i) and (ii) are necessarily misinformation-related, yet their analysis still provides insight into our platform's ability to simulate propagation dynamics within $N_A$.

As shown in Fig 6A, the number of recorded infections within $N_A$ for type (i) posts demonstrates a reasonable correlation with that predicted by the ABM with a Pearson correlation in log-space equal to 0.81 ($p < 0.01$). A positive, albeit not statistically significant, correlation of 0.06 ($p = 0.075$) is observed (Fig 6B) for type (ii) posts. These results suggest most of the

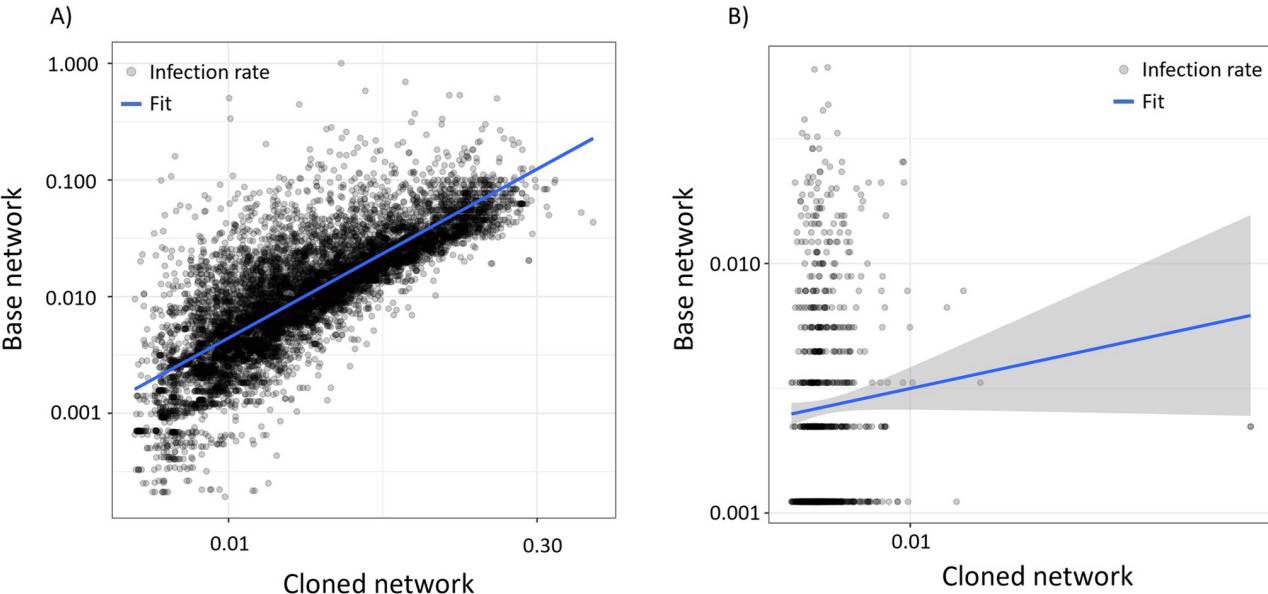

**Fig 6. Comparison of infections in base and cloned networks.** (A) For a set of source posts sampled across all users in our base network, we plot the infection rates extracted from simulating these events within our ABM versus the infection rate measured in the base network ABM (Pearson correlation in log-space equal to 0.81, p < 0.01). Infection rate, which is calculated as number of infections divided by the number of source author followers, is presented to provide a consistent scale across the observations. (B) A similar plot to (A), except all events are sampled from $u_S$ (Pearson correlation in log-space equal to 0.08, p = .075). Since all author-level features are fixed for these events, the visualization conveys the extent to which the ABM can anticipate variations in virality arising solely from post text. In both plots, the blue solid line represents a linear fit to the data, with the bands denoting the 95% confidence intervals of the fit.

variation in virality explained by the ABM is attributable to user-level features; however, the ABM still does demonstrate a degree of text-sensitivity even when user-level features are fixed. For reference, static infection models that do not consider user or text-based features would not display any variation in virality across (i) and (ii) posts. In summary, Fig 6A and 6B demonstrate that post virality varies strongly both across users as well as across posts authored by a single user. This variation is partially reproduced by our dynamic ABM but is largely neglected by more traditional static infection probability frameworks, suggesting the tools explored here may help produce higher fidelity simulations of social network activity. Lastly, while Fig 6 suggests user-level features account for a majority of ABM variation, the infection model architecture can be adjusted to place more or less of an emphasis on text-based features, allowing for balancing of ABM text sensitivity with simulation fidelity.

Aside from understanding how many users a post will infect, understanding how these infections are distributed across online communities is also a key consideration for intervention strategies. To this end, we compare the community infection rates (number of infections / community size) extracted from our cloned and base networks for $T_S$ (Fig 7A), observing an average mean absolute error of 0.065 between the two sets. For comparison, we also ran a static probability version of our ABM that replaced our infection model with a fixed infection rate equal to the average reshare rate of all posts within $N_A$. This baseline achieved a MAE of 0.080, a value roughly 15% larger than our infection model ABM.

In Fig 7B, we also present the community-to-community infection rates within an ABM trial for $T_S$ as a heatmap. The heatmap indicates strong interactions between the two COVID-related communities within $N_A$, as might be expected given the nature of the post. While in our ABM model we can track which member infected another member, there is an ambiguity in the underlying Brandwatch data that makes it unclear whether a user in the base network

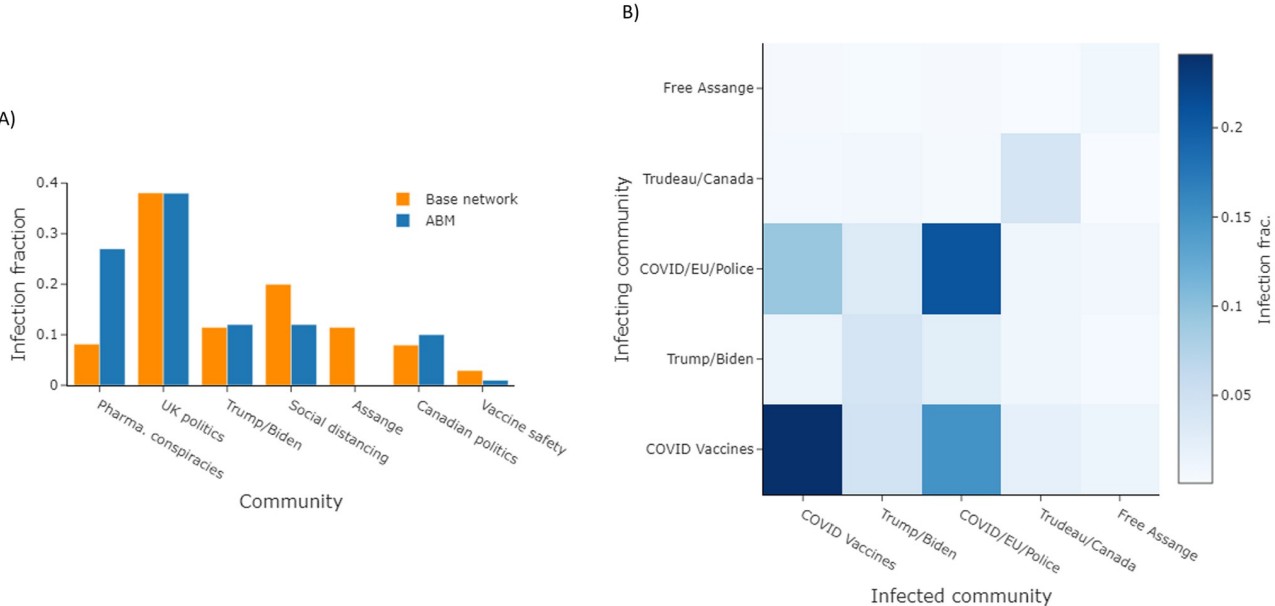

**Fig 7. ABM infections across communities.** (A) A comparison of the distribution of infections rates across communities for $T_S$ between our base network and a simulation of the event with our ABM. (B) A heatmap presenting the community-to-community infection rates recorded when simulating $T_S$ through our ABM, with each grid block representing the fraction of total infections originating from the associated infection pathway.

reacted to $T_S$ or a subsequent retweet of $T_S$ when spreading their infection. Due to this ambiguity, we cannot directly compare infection pathways between the twin networks. However, since understanding community infection pathways is often a starting point within infodemiology [46], we still explore such dynamics to highlight an operational feature of the ABM.

## Countermeasure evaluation

To demonstrate our platform's relevance to countermeasure evaluation, we ran two separate sets of ABM simulations, as discussed below.

**Quarantining of influential individuals.** We first ranked users in descending order of how many infections they caused within our simulation of $T_S$. We then ran a set of simulations where we effectively quarantined varying fractions of the most highly ranked users by rendering them unable to produce infections (account blocking). The results are displayed in Fig 8A. As can be seen in the figure, infection numbers drop precipitously as the number of blocked accounts increases. Social media moderators must carefully weigh the benefits of blocking an individual to prevent harmful content spread on their platform with the costs of stymieing free expression and eroding user trust. Evaluation methods that can estimate how integral different users are to infection spread, and on which topics these users are most influential, may play a role in guiding these risk calculations for moderators.

**Inoculation of dominant infection-spreading communities.** For our second set of simulations, we first identified which community caused the largest number of infections within our ABM simulation of $T_S$. We then simulated an inoculation campaign in this community by reducing all infection probabilities for community members by 20% +/- 2%, a value extracted from research on such campaigns within randomized control trials [21]. The results from these simulations are displayed in Fig 8B. As seen in the figure, the number of infections within the network falls as inoculation rates within the target community increase.

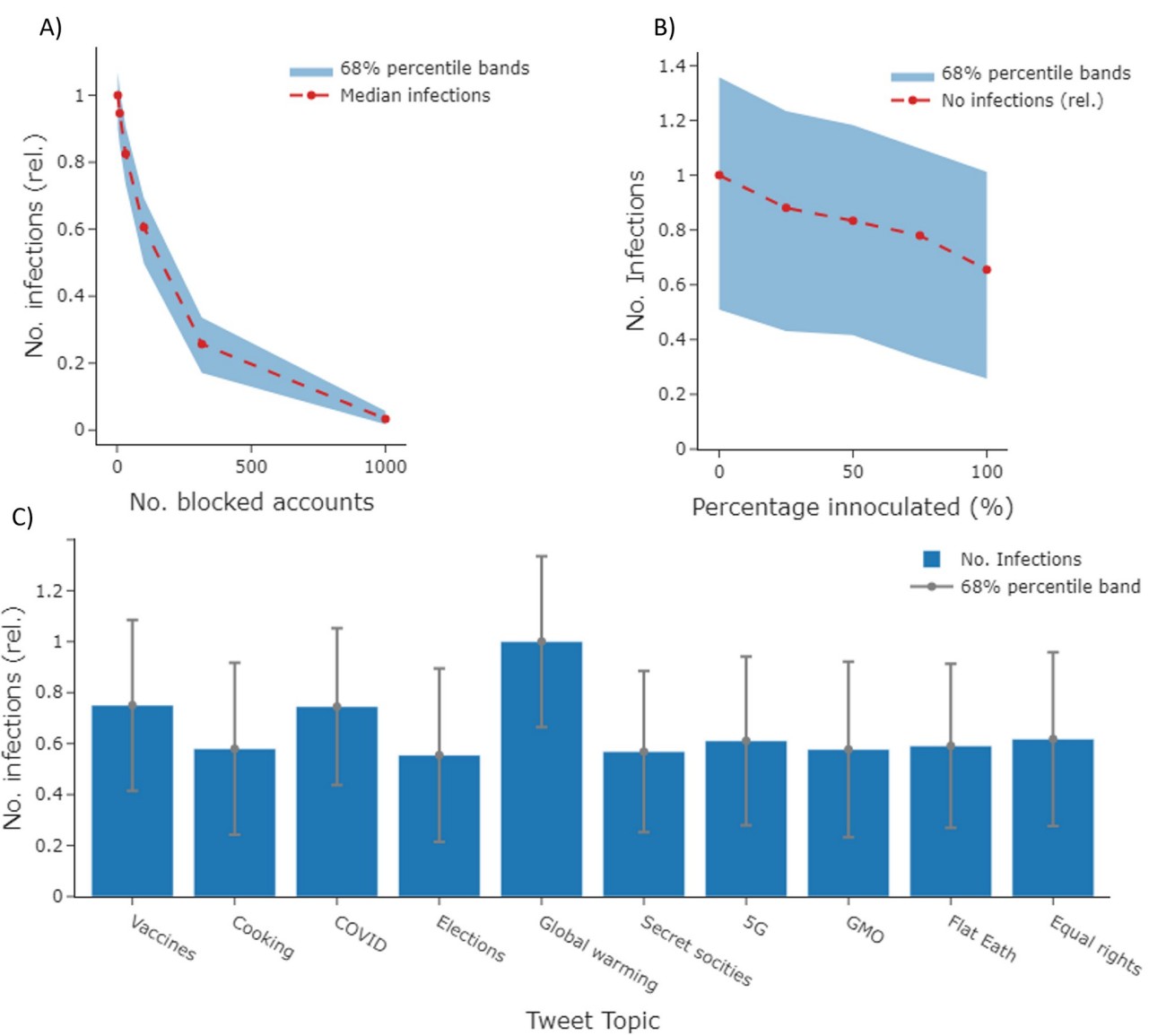

**Fig 8. Countermeasure evaluation and ABM topical sensitivity.** (A) Results for a set of simulations of $T_S$ where we block variable amounts of influential users (x-axis) and measure the corresponding effect on total number of infections within the cloned network (y-axis). We run a base simulation of $T_S$ to identify users that generated the most infections. We then run additional simulations while blocking the top X most influential accounts, where X varies over a range of 0–1000. When a user is blocked in the ABM, they cannot infect other users. (B) We simulate an inoculation campaign within our ABM by running a set of simulations where a variable fraction of users within a community (x-axis) has their output infection probabilities decreased by ~20%. These simulations mimic the effect of inoculation campaigns that reduce the likelihood users will pass on misinformation. As can be seen in the plot, as inoculation fraction decreases, so does the total number of infections recorded within the cloned network (y-axis). The community chosen for inoculation here is the COVID-Vaccines community that generated the most infections within base simulations of $T_S$ (C) We seed our ABM with a set of posts on different common misinformation topics, as well as a baseline post on cooking. We notice large variations in the output infection numbers, indicating information spread within our cloned network is sensitive to topic of discussion. In all three plots, infection numbers are presented on a normalized [0,1] scale.

Inculcation campaigns are being administered through in-person training [21] as well as through digital advertisements [47], channels with differing costs and degrees of effectiveness. With a better understanding of how inoculating different communities will impact overall misinformation spread, public health practitioners can make more strategic decisions about

who to target for inoculation and which inoculation channels to pursue given a finite set of resources.

## Topic sensitivity

Anticipating which misinformation topics may cause the most network activation ahead of time may give social media platform managers and other actors more time to develop tailored mitigation strategies. Another potential use case of our ABM is performing topical red teaming to inform such discussions. To explore this, we ran our ABM using a set of seed posts covering a range of common misinformation topics as well as a non-information topic, cooking, to serve as a reference (S3 Appendix in S1 File). We notice relatively larger mean activations across topics such as global warming, COVID, and vaccines than across topics such as genetically modified organisms (GMO) produce and our baseline topic (cooking). Once again, the variance in infection number across topics demonstrates that our infection model and ABM dynamics are sensitive to topic of discussion, unlike static infection models that are topic-agnostic.

## Discussion

In this work, we present a proof-of-concept system for simulating misinformation spread within online social media networks. We effectively clone a base network of ~10,000 users by producing an agent-based model where each agent is modeled after a user in the base network. Social media histories for each base network user are extracted and transformed into features that are assigned to each agent. Historical misinformation sharing events within the base network are recorded and leveraged to train an infection model that predicts the likelihood that a given social media post will be shared between two network agents. We also deploy LLMs to anticipate how information will be mutated as it propagates through a network. Collectively, the infection model, mutation model, and extracted network relationships ground our cloned network in recorded social media behavior to help anticipate forward-looking misinformation dynamics.

To evaluate our method, we seed our cloned network with a sample of historical posts recorded within the base network and compare infection rates across the network twins, observing positive correlations between the two. Similarly, compared to a static probability ABM baseline, we demonstrate our infection model ABM 15% more accurately anticipates how infections are distributed amongst online communities for a vaccine hesitancy validation event. Lastly, we explore how the ABM may be leveraged for red teaming analysis and for simulating both quarantine-based and vaccination-based misinformation interventions.

However, there are several limitations of this work. Firstly, we evaluated our simulation system by replicating dynamics within a fixed set of X communities known to discuss COVID-19 conspiracy theories. Future work should reapply our framework to a different set of online communities and misinformation topics, either within X or a separate social media platform, to assess the generalizability of our results. Similarly, our ABM was built upon a simple SIR model that neglected more complex user interactions, such as refutation and debunking, known to influence propagation dynamics [48, 49]. Further, we assumed a user exposed all followers to a given retweet; however, X's recommendation algorithm plays a vital role in determining the posts each user views, a mechanism that has been modeled elsewhere [50]. Lastly, as described in the Network selection section, subnetwork generation required removing a set of relationships known to exist within the base network. This process resulted in base and cloned networks with non-identical graph structures, ultimately hampering the fidelity with which the cloned network could replicate base network dynamics.

There are several future directions this work may take. Firstly, in this work, we chose to clone a relatively small social media subnetwork to simplify evaluation of our method. However, it may be desirable to create synthetic networks that are more representative of larger national social media communities to study more widespread misinformation campaigns. Extracting social media histories for all users in these networks is neither practical nor likely necessary. Rather, a small set of recorded histories may be used to generate a much larger synthetic population. Similarly, national social networks can be analyzed and condensed into smaller, more manageable networks that still retain core parent network properties. A combination of community detection at scale, node aggregation [51], and synthetic network generation [52] can be performed to produce networks that are structurally similar to national networks but computationally feasible to both populate with agents and run simulations over.

Secondly, higher dimensional embeddings can be leveraged within the infection model to better capture sensitivities to subtle linguistic features such as tone, emotion, and other stance variables. In line with recent work exploring LLMs for social simulation [53, 54], our binary classification infection model may be replaced by fine-tuned LLMs trained on each community to yield more accurate infection rates and mutation dynamics.

Lastly, the ABM can be modified to process multimodal misinformation content that contains text, video, and image components, which may help extend our framework to other mainstream social media platforms outside of X. While we note that the tools presented here for misinformation mitigation may be adapted by bad-faith actors for misinformation amplification, we hope the open publication of such tools prevents either offensive or defensive actors from gaining a runaway advantage [55]. We believe the work presented here provides a useful step towards more accurately modeling and understanding forward-looking misinformation scenarios as well as developing nuanced mitigation strategies.

## Supporting information

**S1 File.**
(ZIP)

## Acknowledgments

The authors Marek Posard for foundational research design discussions and Melissa Baumann for assistance with graphic design.

## Author Contributions

**Conceptualization:** Prateek Puri, Gabriel Hassler, Sai Katragadda, Anton Shenk.

**Data curation:** Prateek Puri, Gabriel Hassler, Sai Katragadda.

**Formal analysis:** Prateek Puri, Gabriel Hassler, Sai Katragadda, Anton Shenk.

**Funding acquisition:** Prateek Puri.

**Investigation:** Prateek Puri, Sai Katragadda.

**Methodology:** Prateek Puri, Gabriel Hassler, Sai Katragadda, Anton Shenk.

**Project administration:** Prateek Puri, Anton Shenk.

**Resources:** Prateek Puri, Gabriel Hassler, Anton Shenk.

**Software:** Prateek Puri, Gabriel Hassler, Sai Katragadda, Anton Shenk.

**Supervision:** Prateek Puri.

**Validation:** Prateek Puri, Gabriel Hassler, Sai Katragadda, Anton Shenk.

**Visualization:** Prateek Puri, Gabriel Hassler, Sai Katragadda, Anton Shenk.

**Writing – original draft:** Prateek Puri, Gabriel Hassler.

**Writing – review & editing:** Prateek Puri, Sai Katragadda, Anton Shenk.

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
