## [Decision Letter · Decision Letter 0]

15 Mar 2024

PONE-D-24-02968Digital cloning of online social networks for language-sensitive agent-based modeling of misinformation spreadPLOS ONE

Dear Dr. Puri,

Thank you for submitting your manuscript to PLOS ONE. After careful consideration, we feel that it has merit but does not fully meet PLOS ONE’s publication criteria as it currently stands. Therefore, we invite you to submit a revised version of the manuscript that addresses the points raised during the review process.

We look forward to receiving your revised manuscript.

Kind regards,

Matteo Cinelli

Academic Editor

PLOS ONE

4. We notice that your supplementary figure is uploaded with the file type 'Figure'. Please amend the file type to 'Supporting Information'. Please ensure that each Supporting Information file has a legend listed in the manuscript after the references list.

Additional Editor Comments:

The reviewers provided the comments for your paper. Overall, the decision is Major Revision

Reviewers' comments:

Reviewer's Responses to Questions

**Comments to the Author**

1. Is the manuscript technically sound, and do the data support the conclusions?

Reviewer #1: Yes

Reviewer #2: Yes

2. Has the statistical analysis been performed appropriately and rigorously? 

Reviewer #1: Yes

Reviewer #2: Yes

3. Have the authors made all data underlying the findings in their manuscript fully available?

Reviewer #1: No

Reviewer #2: No

4. Is the manuscript presented in an intelligible fashion and written in standard English?

Reviewer #1: Yes

Reviewer #2: Yes

5. Review Comments to the Author

Reviewer #1: The authors provide a valuable exertion towards more accurately modeling and understanding forward-looking misinformation scenarios. I acknowledge the effort of reconstructing the network of users engaged with a given source post. Furthermore, the authors employ a noteworthy number of steps and techniques to account for the tweet-level and user-level features in developing the infection model. Despite various dimensional reductions of embeddings and parameter approximations in estimating probabilities, the model performances are adequate. I appreciated that, as the authors pointed out: "We notice only very slight performance degradation across time, suggesting Period I-II user behavior encoded during the training process remains relevant to user information sharing tendencies for multiple months", displaying a stable value of the FPR through time, despite a degree of overfitting between the training and test sets. I would interpret the latter as a further robustness check of the training procedure, along with its implication from an interpretative point-of-view of stable users' inclination regarding information consumption. This outcome is consistent with the results reported in Fig. 6A and 6B, where the authors anew highlighted that "most of the variation in virality explained by the ABM is attributable to user-level features; however, the ABM still does demonstrate a degree of text sensitivity when user-level features are fixed." I would suggest emphasizing more these outcomes in terms of the model's validity and their implications from an interpretative point of view in misinformation countermeasure evaluation. Finally, I would add a column with the number of posts for each community in Table S1.

Overall, I think this is a very well written and carried-out work, and I recommend this paper for publication.

Reviewer #2: This paper proposes a new model for studying the spreading of misinformation on online social media.

Different from previous works, the authors considered a Machine Learning framework that computes infection probabilities based on communities, users and topicsof discussion. Moreover, they used LLMs to predict how information changes during its transmission (i.e. users sharing posts on Twitter may align differently from the original poster).

The authors also tested their models with real data and provided some possible applications of it.

Although the results are interesting and promising, I think that some aspects need to be revised and some further analysis should be done. Here I provide a detailed list of my concerns:

- the paper uses a "cloned" network obtained starting from a post chosen by the authors. Although they explain why they chose that post, I think that it is necessary to add the analysis and the results for at least one other network to check the robustness and validity of the model;

- Line 119-120, the authors say they selected the 10000 most active users, measured with the in-degree of the node (i.e. how many times the user shares a post of another). My concern here is the following: consider a network in which node v has a high in-degree, but its neighbour is composed of user with low in-degree (similar to a star pointing to v). Thus, they are deleted in the filtered network G. In this way, node v is part of G, but its "activity" is highly underestimated because all its neighbour neighbours have disappeared. This is a possible problematic case that can arise but, in general, I think that the activity should be measured differently. Maybe both the in- and out-degree of nodes should be considered to measure activity. Moreover, I suggest to add some simple statistics of the network (edge density, degree distribution, etc.);

- Line 136-137, the authors say that in Fig1A the edge thickness is proportional to the number of follower-followee between the two communities. However, in Line 121-122 it is written they didn't extract the relation followee-follower.

Could you explain better?

- Line 168-189 (SEI Model Pseudocode), I found it hard to read due to the difficult-to-read notation. Moreover, it seems that some symbols are not defined. In particular:

1. Line 173: what is f(j)? Is t_j the time at which a user becomes infected? In case, please specify it better;

2. Line 177: I think you could simply say S(f_j) = S;

3. Line 179: (at least for me) it's very hard to read and interpret X(p = IP). Since it is a Bernoulli random variable,

please use a simpler notation;

In general, please check you defined everything you mention in the pseudocode.

- Line 193: The authors say that \\Delta is an arbitrary random variable, but I didn't find what type of distribution they use for it in the simulations. Please add some more details;

- Line 246: maybe you can add a citation to your sentence;

- Line 318-319: the authors mention correlation values that are not shown. Please add them in the Figure or in a separate table, together with their significance (e.g. p-values). Moreover, please specify what coefficient did you use and if correlations are computed on logged values instead of original;

- At the end of the discussion section, please add a paragraph about the limitations of the work. Although you mentioned some of them throughout the paper, I think it could be useful to provide a summary in the end.

6. PLOS authors have the option to publish the peer review history of their article (what does this mean?). If published, this will include your full peer review and any attached files.

Reviewer #1: No

Reviewer #2: No

---

## [Author Response · Author response to Decision Letter 0]

26 Apr 2024

We have performed additional analysis, updated manuscript text, and updated our manuscript figures in response to the reviewer/editor comments on our initial submission. In the attached Rebuttal Letter document, we respond to each comment individually by highlighting the specific revisions made in response to each.

---

## [Decision Letter · Decision Letter 1]

21 May 2024

Digital cloning of online social networks for language-sensitive agent-based modeling of misinformation spread

PONE-D-24-02968R1

Dear Dr. Prateek Puri,

We’re pleased to inform you that your manuscript has been judged scientifically suitable for publication and will be formally accepted for publication once it meets all outstanding technical requirements.

Kind regards,

Matteo Cinelli

Academic Editor

PLOS ONE

Additional Editor Comments (optional):

Dear Authors,

the reviewers recommended your article for publication.

Reviewers' comments:

Reviewer's Responses to Questions

**Comments to the Author**

1. If the authors have adequately addressed your comments raised in a previous round of review and you feel that this manuscript is now acceptable for publication, you may indicate that here to bypass the “Comments to the Author” section, enter your conflict of interest statement in the “Confidential to Editor” section, and submit your "Accept" recommendation.

Reviewer #1: All comments have been addressed

Reviewer #2: All comments have been addressed

2. Is the manuscript technically sound, and do the data support the conclusions?

Reviewer #1: Yes

Reviewer #2: Yes

3. Has the statistical analysis been performed appropriately and rigorously? 

Reviewer #1: Yes

Reviewer #2: Yes

4. Have the authors made all data underlying the findings in their manuscript fully available?

Reviewer #1: No

Reviewer #2: No

5. Is the manuscript presented in an intelligible fashion and written in standard English?

Reviewer #1: Yes

Reviewer #2: Yes

6. Review Comments to the Author

Reviewer #1: The authors addressed all my comments adequately. I deem there are no other issues, and I recommend this paper for publication.

Reviewer #2: Thanks to the authors for having addressed all my previous comments.

I think that the work has been greatly improved and it is ready to be published.

7. PLOS authors have the option to publish the peer review history of their article (what does this mean?). If published, this will include your full peer review and any attached files.

Reviewer #1: No

Reviewer #2: No

---

## [Editor Report · Acceptance letter]

27 May 2024

PONE-D-24-02968R1 

PLOS ONE

Dear Dr. Puri, 

I'm pleased to inform you that your manuscript has been deemed suitable for publication in PLOS ONE. Congratulations! Your manuscript is now being handed over to our production team.

Kind regards, 

on behalf of

Dr. Matteo Cinelli 

Academic Editor

PLOS ONE